# Experimental Evaluation of a Hybrid Sensory Feedback System for Haptic and Kinaesthetic Perception in Hand Prostheses

**DOI:** 10.3390/s23208492

**Published:** 2023-10-16

**Authors:** Emre Sariyildiz, Fergus Hanss, Hao Zhou, Manish Sreenivasa, Lucy Armitage, Rahim Mutlu, Gursel Alici

**Affiliations:** 1School of Mechanical, Materials, Mechatronic and Biomedical Engineering, University of Wollongong, Wollongong, NSW 2522, Australia; hzhou@uow.edu.au (H.Z.); manishs@uow.edu.au (M.S.); alucy@uow.edu.au (L.A.); gursel@uow.edu.au (G.A.); 2Orora, 109 Burwood Rd., Hawthorn, VIC 3122, Australia; fergushanss@gmail.com; 3Faculty of Engineering and Information Sciences, University of Wollongong in Dubai, Dubai P.O. Box 20183, United Arab Emirates; rmutlu@uow.edu.au; 4The Intelligent Robotics & Autonomous Systems Co. (iR@SC), Shellharbour, NSW 2529, Australia

**Keywords:** haptics, proprioception, prosthesis, robotic hand, sensory feedback

## Abstract

This study proposes a new hybrid multi-modal sensory feedback system for prosthetic hands that can provide not only haptic and proprioceptive feedback but also facilitate object recognition without the aid of vision. Modality-matched haptic perception was provided using a mechanotactile feedback system that can proportionally apply the gripping force through the use of a force controller. A vibrotactile feedback system was also employed to distinguish four discrete grip positions of the prosthetic hand. The system performance was evaluated with a total of 32 participants in three different experiments (i) haptic feedback, (ii) proprioceptive feedback and (iii) object recognition with hybrid haptic-proprioceptive feedback. The results from the haptic feedback experiment showed that the participants’ ability to accurately perceive applied force depended on the amount of force applied. As the feedback force was increased, the participants tended to underestimate the force levels, with a decrease in the percentage of force estimation. Of the three arm locations (forearm volar, forearm ventral and bicep), and two muscle states (relaxed and tensed) tested, the highest accuracy was obtained for the bicep location in the relaxed state. The results from the proprioceptive feedback experiment showed that participants could very accurately identify four different grip positions of the hand prosthesis (i.e., open hand, wide grip, narrow grip, and closed hand) without a single case of misidentification. In experiment 3, participants could identify objects with different shapes and stiffness with an overall high success rate of 90.5% across all combinations of location and muscle state. The feedback location and muscle state did not have a significant effect on object recognition accuracy. Overall, our study results indicate that the hybrid feedback system may be a very effective way to enrich a prosthetic hand user’s experience of the stiffness and shape of commonly manipulated objects.

## 1. Introduction

The loss of upper limbs is a traumatic and life changing event, and can create barriers to occupational and social participation, as well as the capability for daily living activities [1]. Significant efforts have been made to develop functional and dexterous hand prostheses to restore normality for people with upper limb loss [2,3,4,5]. Thanks to the rapid developments in robotics, electric prosthetic hands can provide much better performance than their body-powered (BP) counterparts [6,7]. A key factor enabling reliable operation of an electric prosthetic hand is an effective closed-loop Human Machine Interface (HMI). Typically, this includes a myo-based user intention recognition system and a sensory feedback system. Though the former has been the focus of significant research [6], the latter is equally critical to provide quality of life to prosthetic hand users [7,8,9]. 

The open-loop control adopted by most commercial hands can be ineffective since visual feedback alone may not be enough to support fine control of object grasping [10]. With sensory feedback integrated, the closed-loop control has been shown to enhance the grasping quality and efficiency for a prosthetic hand user [11,12,13]. Bouteraa et al. reported the closed-loop control based on a fuzzy logic controller with hybrid biofeedback to improve robotic manipulation and grasping capabilities [14]. In addition, with the restoration of proprioception and the ‘feeling’ of touch, sensory feedback can greatly improve the prosthetic embodiment and contribute to the reduction in high rejection rates (~40%) of hand prostheses [7]. 

There is limited integration of sensory feedback into commercial prosthetic hands. Most feedback systems in the literature are still within the laboratory testing phase and not mature enough for clinical use. These systems can be broadly grouped into two categories, invasive and non-invasive methods [7]. Invasive methods apply stimulation directly to the nerves, which generally requires surgical procedures to access relevant nerves. On the other hand, non-invasive methods involve cutaneous stimulation of another body part when the prosthetic device performs a gesture or grasps an object [7]. Non-invasive methods are currently preferred due to concerns about long-term effects and the safety of invasive methods [15]. 

Two major concerns arise for non-invasive feedback: stimulation modalities and locations. Having both modality-matched and somatotopical feedback is ideal to achieve natural sensations, and leads to easier understanding and quick responses for the user. 

Various cutaneous stimulation methods have been developed for sensory feedback, including vibrotactile, mechanotactile, electrotactile, and a hybrid combination of these [7,8]. 

Vibrotactile feedback, with small vibrating motors attached to the skin, is one of the most common methods due to its ease of use (e.g., lightweight, low cost, low footprint, low energy consumption, etc.) [16]. It is usually used to effectively provide both kinaesthetic and haptic senses to users [17,18]. Vibrotactile feedback has also been demonstrated to have the capacity to improve users’ sensorimotor ability to accomplish delicate grip tasks [19]. Additionally, it is reported that users find vibrotactile feedback more comfortable than alternatives such as electrotactile feedback [20]. However, vibrotactile feedback is limited by its slower response time (up to 400 ms) with the feedback only being detectable in a limited range of vibrational frequencies [7]. 

Mechanotactile feedback involves applying a mechanical force to the user, usually in the form of pressure or skin stretch, or a combination of the two. It has been used to effectively communicate kinaesthetic and haptic information to users [21,22,23,24]. One of the major advantages is the possibility of applying continuous feedback to users, which is a greatly desired feedback characteristic as users can obtain more information than other alternatives such as vibrotactile arrays. Another advantage is its modality-matching sense for the pressure/force exerted on the prosthetic hand during grasping or touching. Using modality matching can greatly reduce cognitive loads for the users and make the sensory feedback easier to interpolate [24,25]. 

Electrotactile feedback produces cutaneous stimulation by generating a low electric current through the skin via a pair of electrodes. It is usually used to provide pressure and slip feedback. Since no mechanical or moving parts are required, it can be advantageous in both space and energy consumption. Additionally, electrotactile feedback can be controlled by altering the intensity, pulse width and frequency of the current. The major drawback is the modality difference. The electrotactile sensation is generally felt as a tingling, which is distinct from the sensation of pressure, making it unnatural for force feedback. In addition, it is more susceptible to skin conditions and electrode displacements. Calibration is also more complicated since all the parameters of electric current cause large effects [7]. 

Biological skin consists of different types of sensory receptors, allowing a human hand to receive all kinds of information, e.g., proprioception, object properties, etc., to help gestures and grasps. Clearly, a single-modal sensory feedback system cannot deliver many stimuli simultaneously. For example, Takahashi et al. reported the comparison between hybrid and single-modal feedback (stimulators attached on a biological hand) for object hardness discrimination and roughness discrimination, respectively [26]. It was found that hybrid feedback performed better in both discrimination tests. An example of a hybrid feedback system was reported by Vargas et al. [27], in which vibrotactile and electrotactile feedback were combined to test, on 9 able-bodied subjects and 1 amputee, the recognition performance of grasped objects’ size and stiffness. The upper arms were selected as the stimulation sites for all the participants. The hybrid feedback allowed simultaneous identification of various object properties to improve grasping quality for prosthetic hand users. Another example was reported by Jimenez et al. [28], in which a prosthetic finger was able to sense temperature, vibration and grasping force. All three types of sensory feedback were sent to the users via modality-matched stimulations (e.g., using pressure for force feedback, using heating/cooling for temperature feedback, etc.) and the users accurately recognized the mass, temperature and roughness of the tested objects. The matched modality usually renders the feedback more intuitive, leading to less cognitive burden for a user. Therefore, hybrid feedback systems with modality-matched stimulations have been drawing increasing attention recently. Several other noticeable studies on hybrid feedback were introduced in a review by Svensson et al. [25]. To restore the sensory function of a hand, hybrid stimulation is promising due to its capability of delivering multi-modal feedback. However, limited exploration into hybrid feedback systems has been conducted when compared to other feedback types. There is a need to further investigate the efficacy of hybrid feedback and grow a broader understanding about how to improve/optimize its configuration (e.g., modality selection, co-located or off-located placement, cooperation between multiple modalities, etc.) based on the required multi-modal information to be delivered. 

Besides stimulation modality, the placement of stimulators also plays a critical role. For non-invasive sensory feedback, somatotopical feedback can be achieved by stimulating the skin of the phantom hand. For a transradial prosthetic hand user, these locations are often near the distal end of the amputated forearm, where it is difficult to attach the stimulators. In addition, the sensation of phantom hand varies greatly among end users, with some amputees having no sensation at all. Compared with the forearm, the upper arm is an attractive option since the usable area is much larger and there would be minimal interference with the intention recognition system, which generally uses surface electromyography (sEMG) electrodes on the forearm. 

There is limited quantitative information available on the optimal placement of stimulation. Often, no reasoning is given within the design of the feedback systems for the final placement of the stimulators. Some work has reported ideal placement of vibrating actuators [29,30]. Stephens-Fripp et al. tested the just notable difference (JNS) at two locations with a mechanotactile system [31]. No statistical difference between the upper and lower arm was found in terms of sensitivity and recognition of mechanotactile feedback. To the best of our knowledge, very limited work has been conducted to directly and comprehensively compare the performance between stimulation sites in terms of force-level identification. 

In this work, a hybrid sensory feedback system, consisting of vibrotactile and mechanotactile feedback, is proposed and investigated in terms of the capability of delivering the information about gripping force and the stiffness of the object in contact simultaneously to improve grasp safety and quality. This study also comprehensively investigates and compares the performance of the forearm and upper arm as the stimulation sites for force-level identification of mechanotactile feedback at different muscle conditions, relaxed and tensed states. It shows that (i) the force level of the sensory feedback system has a significant effect on the accuracy of haptic perception regardless of the feedback location and muscle states, (ii) the accuracy of haptic perception is higher at the bicep location in the relaxed state, (iii) the location of haptic feedback does not have a statistically significant effect in the tensed state, and (iv) multi-modal feedback can be employed to accurately identify force and object properties simultaneously through haptic perception. These results provide a solid reference to guide the selection of stimulation modalities and optimal stimulation site for applying hybrid feedback, which plays a critical role in developing a sensory feedback system and consequently the closed-loop control of a prosthetic hand, providing safe and efficient grasps, as well as making the users ‘feel’ again. 

The rest of the paper is organized as follows. Section 2 introduces the hybrid sensory feedback system that provides haptic and kinaesthetic perception in hand prostheses. Section 3 explains the experimental methodology in detail. Section 4 represents the experimental results and analysis. The paper ends with discussions and conclusions given in Section 4 and Section 5.

## 2. A Hybrid Sensory Feedback System for Hand Prostheses

The experimental system was developed by employing a soft prosthetic hand illustrated in Figure 1 and a hybrid sensory feedback system illustrated in Figure 2. The motion controllers of the prosthetic hand and hybrid sensory feedback system were implemented using MATLAB Simulink Real–Time, Quanser’s Q8–USB data acquisition card, a Maxon DC motor with an encoder of 10,000 ppr and a reduction gear of 1:30, and Speed studio mini vibrating motors. The sampling time of the real-time control system was 1 ms. The gripping force and haptic feedback force exerted on the arm were estimated using the Force Sensitive Resistor (FSR) sensors illustrated in Figure 1 and Figure 2. 

### 2.1. Soft Prosthetic Hand

The dexterity of a human hand can be approximated as 27 degrees of freedom (DoF) with 5 DoF dedicated to the thumb [32]. The complex mechanical structure and high DoF of a human hand still present a great challenge in the design and implementation of a robotic hand, particularly in the field of robotic prostheses. Such dexterity and versatility in a robotic hand can be practically achieved by employing a complex mechanical design with individual active joints controlled independently. This approach, however, dramatically increases the size and weight of the robotic hand, both of which are barriers to everyday use. It is therefore essential to build a compact and effective robotic hand prosthesis that can be used to perform various daily activities by hand amputees while improving the acceptance of the prosthetic hand with integrated sensory feedback. 

In previous work, we developed a soft robotic hand with compliant joints [4,33,34,35], reducing mechanism complexity, while being able to perform the most common daily hand gestures: grasp, two-finger grip (i.e., pinch), and envelop grip. In addition to functionality, the soft robotic hand provides safety in human–robot interaction with its intrinsically compliant mechanical structure [36,37]. Yong et al. reported that a person either right or left handed uses their fingers in their daily lives at different rates; utilization of the thumb is 37.6%, index finger and middle finger account for 19.7% and 21.2% respectively, and the rate of the ring finger is 5.6% and little finger is 2.4% [38]. The rate of use of the thumb is the highest among other fingers, which is followed by similar rates of index and middle fingers. With these findings in mind, the soft robotic hand used in this study is actuated by three DC motors providing three DoF; the thumb, index finger and the other three fingers are actuated with a single tendon-cable (see Figure 1) [33,34,35]. The tendon-cable actuation provides the ability to flex soft fingers with multiple DoF using a single actuator for each finger, resulting in an underactuated system with reduced complexity. Although this method does come with the sacrifice of some control, the overall reduction in weight and control complexity is a desirable characteristic of this system. 

To estimate the gripping force of the prosthetic hand, we placed an FSR sensor on the thumb finger as shown in Figure 1b. This estimated force was used to generate the force reference signal of the haptic feedback system explained in Section 2.2. 

### 2.2. Hybrid Sensory Feedback System

To provide multiple modes of proprioceptive sensory information (e.g., touch and movement senses) to the user, we developed a hybrid sensory feedback device illustrated in Figure 2. In addition to relaying haptic perception (Figure 2a) and kinaesthetic sense (Figure 2b), the hybrid sensory feedback system is intended to enable users to perform dexterous tasks such as object detection based on its shape and stiffness. While a single location on the left arm was used for the kinaesthetic feedback, the haptic feedback was applied to three different locations on the right arm: the volar surface of the forearm (F1), the ventral surface of the forearm (F2), and the bicep (B) as shown in Figure 2c. The overall experimental setup, including the soft prosthetic hand and the hybrid sensory feedback system, is illustrated in Figure 2d, while the close-up views of the sensory feedback systems are illustrated in Figure 2e. 

The proposed haptic feedback system comprises two main components: (i) a force measurement system placed on the soft prosthetic hand, and (ii) an active force controller implemented in the sensory feedback system. While the force measurement system was used to identify the grasping force of the prosthetic hand, the active force control system allowed us to regulate the force exerted on the arm in different locations and muscle states in the haptic feedback and object recognition experiments. 

When the soft prosthetic hand held an object, the force exerted on the robotic hand was measured using an FSR sensor. It is noted that we can further reduce the cost by estimating the grasping force information via force observers [39,40]. The estimated force by the FSR sensor on the prosthetic hand was used as the reference of the active force controller in the sensory feedback system. In other words, we generated the haptic feedback signal using the identical force of the prosthetic hand in force feedback experiments. However, we can simply generate different force references, e.g., we can scale up force references to improve the sensitivity of physical interaction tasks as proposed in teleoperation systems [40,41]. The active force control system was implemented using another FSR sensor in the sensory feedback system shown in Figure 2e and a proportional force feedback controller proposed in [39,42]. The proportional force feedback controller regulated the force exerted on the arm by suppressing disturbances in different locations and muscle states. 

The kinaesthetic feedback system, illustrated in Figure 2b, was developed by using four vibrating motors sequentially placed on an arm strap, i.e., a vibrotactile array (see Figure 2b,e). The information of various hand movements can be transmitted to the users by adopting different activation methods of the vibrotactile array, such as tactile apparent movement [43] and tactile encoding schemes [44]. For the sake of brevity, this study only considered the cylindrical and pinch grasping motions of the prosthetic hand. These motions are essential to perform various daily activities such as buttoning a shirt or holding a cup [45]. The motion range of the robotic hand was divided into four discrete states: (i) open-hand, (ii) wide grip, (iii) narrow grip and (iv) closed-hand. The vibrating motors were sequentially activated when the state of the robot hand changed from open to closed hand. While the static response of the vibrating motors provided position information, the users could sense the speed of the grasping motion by the dynamic response of the vibrotactile array. 

## 3. Experimental Methodology

### 3.1. Participants

A total of 32 participants took part in three experiments: experiment 1 with 10 participants (8 males, 2 females, age 22.8 ± 1.23 years and height 182 ± 8.01 cm), experiment 2 with 11 participants (8 males, 3 females, age 26.09 ± 6.13 years and height 180.90 ± 6.05 cm), and experiment 3 with 11 participants (8 males, 3 females, age 26.09 ± 6.13 years and height 180.90 ± 6.05 cm). Participants volunteered to take part in the experiment and were recruited by convenience sampling from a pool of students and staff at the University of Wollongong, Australia. Participants had normal or corrected-to-normal vision and reported to be in good general health. Two participants wore glasses. The experiments were conducted in accordance with the standards of the Declaration of Helsinki (rev. 2013), with formal approval of the Human Research Ethics committee at the University of Wollongong, Australia (ID No: 12402, Protocol No: 2022/364). 

The experiments were run in separate sessions spread over several days, with each experiment taking up to 35 min per participant. Prior to each session/experiment, a 5 min standardized training session was provided to make participants familiar with the experimental protocol.

### 3.2. Experimental Protocol

In all experiments, participants were seated comfortably at a desk and were able to rest their arms in front of them (Figure 2d). Their arms were placed on a bench and the operator positioned the required devices on the target location(s). Once positioning was finalized, the participant’s right arm was held in place using straps to prevent unintended movement during the experiment. At all times, a cut-off switch was available to both the user and operator as a safety precaution. 

Participants were not able to visualize any of the prosthetics or control systems used to run the experiment when they were seated. Three locations were identified on the right arm of each participant: the central portion of the volar surface of the forearm (F1), the central portion of the ventral surface of the forearm (F2), and the muscle belly of the bicep (B) (Figure 2c). These locations were used as the application points for the force feedback. The vibrotactile feedback band was applied to the central portion of the left forearm. 

Preliminary threshold testing was performed to ensure that each participant was only subjected to forces within their self-identified comfortable range. Forces in 1 N increments up to a maximum of 10 N or to the onset of discomfort were applied to each location on each participant, and they were asked whether they were comfortable. Subsequent testing was limited to a force range 2 N below the force threshold where any discomfort was experienced. Across all participants, the maximum comfortable force limit was identified to be 8.5 ± 1.2 N, and therefore the applied experimental forces were limited to 7 N. 

**Experiment 1 (EXP1):** Experiment 1 aimed to assess the accuracy of force perception of users at different anatomical locations and for different levels of muscle contraction (relaxed (R) or tensed (T)). EXP1 followed a 3 location (F1, F2, B) × 2 muscle state (R, T) × 7 force level (FL1 to FL7) design. In the training phase, participants were presented with different force levels and asked to take note of the sensation. They were told that they would be tasked to try and identify these force levels in the experiment. During the experiment, at the beginning of each trial, participants were verbally instructed to match a randomly chosen force level (from FL1 to FL7). Subsequently, a linear force ramp was applied, and participants had to halt the force application using a button when they perceived that the previously communicated force level had been reached. Due to the impracticality of switching locations between trials, all trials at one location were recorded together; however, the order of locations was randomized between participants. To minimize fatigue, muscle states were tested alternately. We measured the signed error between the force level required for each trial, and that indicated by the participant. 

**Experiment 2 (EXP2):** Experiment 2 aimed to assess whether participants could identify the hand position based on vibrotactile feedback. The vibrotactile cuff (Figure 2b) was attached to the thickest part of each participants’ left forearm. In the training phase, each of the four motors in the cuff were activated sequentially in ascending and descending order so that participants could familiarize themselves with the position and pattern of vibrations. During the experiment, the full grip range of the prosthesis was represented by activating the motors randomly to represent one of the four discrete positions (25%, 50%, 75% and 100%) or by not activating any motors to indicate 0% grip. Participants were then asked to verbally state which motor they believe had been activated. Each motor was tested 3 times in random order. We measured the success rate (binary, 0 or 1) depending on whether the participant accurately identified the motor being actuated. 

**Experiment 3 (EXP3):** Experiment 3 aimed to assess the ability of the participants to differentiate between items of different stiffness using the hybrid feedback system to represent both the force and position information associated with the stiffness of an object. Four possible material types were presented: empty grasp E, low stiffness foam block (L), medium stiffness packaging foam (M), and high stiffness wooden block (H). In the training phase, participants were made familiar with the feedback from each material type. Similar to EXP1, these scenarios were tested across muscle states and locations, giving a 3 location (F1, F2, B) × 2 muscle state (R, T) × 4 material (E, L, M, H) design. Each combination was tested 3 times. We measured the success rate (binary, 0 or 1) depending on whether the participant accurately identified the material being grasped.

### 3.3. Analysis

Recorded experimental data were processed in Excel 365 (Microsoft Version 2022) and MATLAB 2022b (The MathWorks, Inc., Natick, MA, USA) and prepared for further statistical analysis in SPSS v28 (IBM). Signed errors from EXP1 were analyzed using an ANOVA with factors location (F1, F2, B), muscle state (R, T), and force level (L1, L2, L3, L4, L5, L6, L7). Trials were averaged across repetitions. For EXP2, we report the raw success rates across all trials and participants. The results of EXP3 were analyzed using Multinomial Logistic Regression with the success rate as the categorical dependent variable, and location (F1, F2, B), muscle state (R, T), material (E, L, M, H) and repetitions as the categorical independent variables. The significance level was set to 0.05. In cases of violations of sphericity, we applied the Greenhouse–Geisser correction and reported the corrected degrees of freedom.

## 4. Results and Discussions

Across all the training and experimental sessions, the hybrid feedback system worked effectively, and participants were able to use it as intended. None of the participants reported discomfort nor did they break off the experimental sessions due to any issues when the experiments were conducted within users’ comfortable force levels. The emergency cut-off (under participant control) was never activated. 

In EXP1, we observed that the average signed error between the requested force level and the one indicated by the participants significantly changed with different force levels (Figure 3), F(1.859, 16.727) = 6.408, *p* = 0.01. It is noted that the ANOVA’s F statistic is reported for the corrected degrees of freedom, and *p* denotes the significance level. Participants showed a positive error (indicating an overestimation) for lower force levels, and a negative error (indicating an underestimation) for higher force levels. This trend was generally followed across locations and muscle states, with some observable differences between relaxed and tensed states, especially for forearm ventral and bicep locations. To further investigate this, we conducted separate ANOVAs for the two muscle states. We found that there was a significant effect of location for the relaxed state (F(2, 18) = 4.187, *p* = 0.032), with smaller errors at the bicep location. 

This experiment clearly shows that the force level of the sensory feedback system has a significant impact on haptic perception regardless of the feedback location and muscle states. As the haptic feedback increased, the participants tended to underestimate the actual force applied to three different locations on their arms, F1, F2 and B. Moreover, the percentage of the force estimation error tended to decrease as the applied force was increased. This aligns with similar studies that apply haptic feedback to different locations on the human body, e.g., toes in [46]. It is noted that force levels are directly related to the mechanical design of the haptic feedback system shown in Figure 2a,e. Further studies should be conducted to optimize the mechanical design of the haptic feedback system. 

While the force estimation error was limited in all locations F1, F2 and B, we observed smaller errors when haptic feedback was applied to location B in the relaxed state. This is a promising result for prosthetic hand applications. While the robotic hand can be controlled through sEMG sensors placed on the forearm, sensory feedback can be provided using location B with minimal interference with an intention recognition system [31]. However, while the force estimation errors were limited in all locations, we did not observe a statistically significant result in the tensed state. Bicep activation and deactivation are likely to happen during many upper limb activities. Therefore, further studies should be conducted to evaluate the effect of changing muscle states on this feedback modality. 

In EXP2, we observed that all participants were able to successfully identify the vibration pattern presented in the proprioception experiment for all trials, without a single case of misidentification. No further analysis was conducted for EXP2. 

Compared to the haptic feedback system, which has a continuous state between 0 N and 7 N, the proprioception feedback system has only four discrete states: open hand, wide grip, narrow grip, and closed hand. This experiment clearly shows that all participants could easily adapt to discrete sensory feedback with limited states within a short training time. Despite high accuracy, the limited states lead to several drawbacks in practice, such as low resolution in prosthetic hand motion and limited dexterity. The interplay between continuous and discrete sensory feedback systems, and the implications for closed-loop feedback in prostheses should be further investigated to clarify their merits and demerits in practice. 

As presented in Table 1, EXP3 showed an overall high success rate of 90.5% across all combinations of location, muscle states and material. There were no statistically significant effects of the conditions tested on the success rate. We observed that while most participants did well to identify the different materials, some (e.g., participants 6, 8) had more misidentifications (Table 1). Interestingly, none of the participants misidentified the empty condition (i.e., when no material presented) and across all trials, the most misidentified material was the medium stiffness packaging foam (n = 29) followed by high stiffness wooden block (n = 24), where n represents the number of misidentifications in EXP 3. 

These results show that the proposed multi-modal sensory feedback system allowed participants to identify various objects with high accuracy using haptic and proprioception feedback without the aid of vision. This could significantly improve the functionality of robotic hand prostheses. While highly accurate object identification results are very promising, more effort should be expended to understand the performance differences observed in experiments, e.g., the difference between participant 1 and participant 6 in Table 1. Moreover, further studies should be conducted to evaluate the performance of the hybrid sensory feedback system during functional activities, not standardized and static upper limb positions.

## 5. Conclusions

In this paper, we presented a novel hybrid multimodal sensory feedback system using a mechanotactile feedback system for haptic perception and a vibrotactile feedback system for proprioception. In addition to force and position recognition, the multimodal-sensory feedback enabled participants to recognize objects with different shapes and stiffness without the aid of vision. Our results carry important implications for feedback-based control of prostheses and can help significantly improve the functionality of robotic prostheses. 

The experiments conducted with a total of 32 participants have shown that the force level has a significant effect on haptic perception. Although underestimation becomes more dominant, the percentage of force estimation error tends to decrease as the force level increases. The most accurate haptic perception results were obtained when the force feedback was applied to biceps in a relaxed state. This result is promising for prosthetic hand control because a sensory feedback system on the upper arm provides great flexibility for locating the intention recognition systems on the forearm. In future work, it would be interesting to further understand the reasons behind performance differences between muscle states and force levels, as well as during dynamic upper limb activity. Highly accurate haptic and proprioceptive feedback systems enabled participants to identify the objects with an overall success rate over 90%. However, we observed significant performance differences between participants, spanning from ~42% to 100% success rates. Another interesting avenue for future research could be to investigate user differences, in particular on people with upper limb amputation, and develop methods to improve the accuracy of the object recognition for all users.

## Figures and Tables

**Figure 1 sensors-23-08492-f001:**
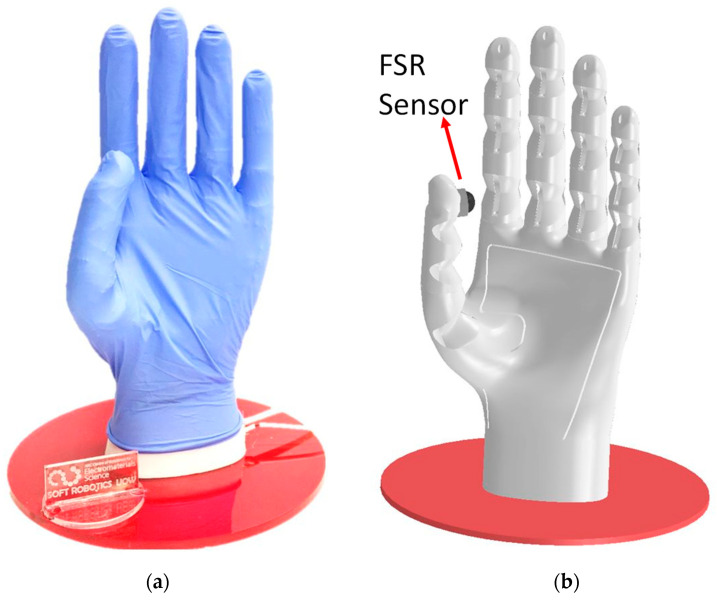
Soft prosthetic hand. (**a**) Prototype. (**b**) CAD model.

**Figure 2 sensors-23-08492-f002:**
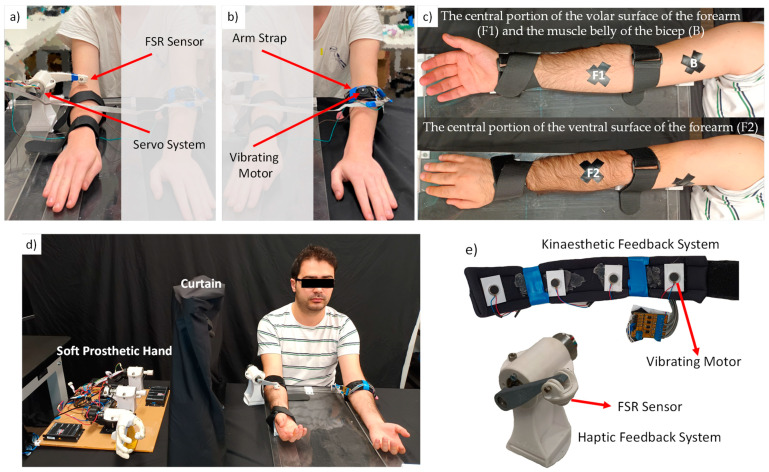
Experimental setup of the hybrid sensory feedback system. (**a**) Haptic feedback system. (**b**) Kinaesthetic feedback system. (**c**) Haptic feedback locations on the arm. (**d**) A subject performing the experiment. (**e**) Close-up views of the sensory feedback systems.

**Figure 3 sensors-23-08492-f003:**
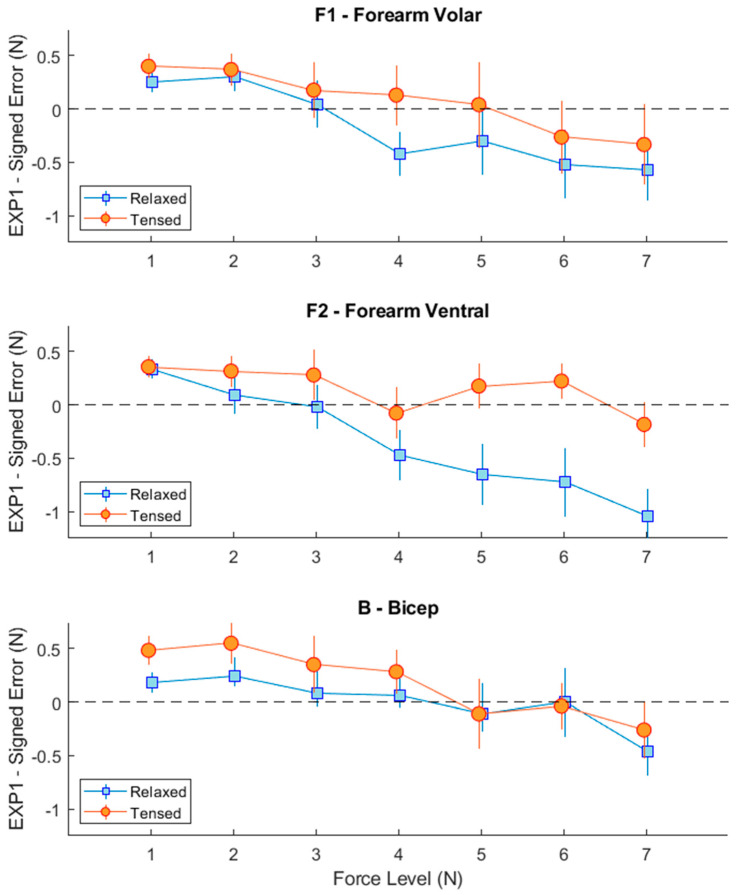
Signed error between applied force level and that indicated by participants in EXP1, compared across relaxed and tensed muscle states. Error bars indicate standard errors of the mean across participants.

**Table 1 sensors-23-08492-t001:** Object identification: Table 1 indicates the percentage success rates for all locations and muscle states. Numbers in brackets indicate the materials presented in the trials that were wrongly identified (out of 3 repetitions per material). P = participant, E = empty, L = low stiffness, M = medium stiffness, H = high stiffness.

	Relaxed State			Tensed State		
P	F1	F2	B	F1	F2	B
**1**	100%	100%	100%	100%	100%	100%
**2**	100%	83.3% (0E,1L, 1M,0H)	100%	91.7% (0E,0L, 1M,0H)	100%	100%
**3**	100%	91.7% (0E,0L, 0M,1H)	100%	100%	100%	100%
**4**	100%	75% (0E,1L, 1M,1H)	100%	91.7% (0E,0L, 0M,1H)	100%	83.3% (0E,0L, 2M,0H)
**5**	100%	100%	100%	100%	91.7% (0E,0L, 0M,1H)	100%
**6**	58.3% (0E,1L, 2M,2H)	83.3% (0E,1L, 1M,0H)	91.7% (0E,1L, 0M,0H)	66.7% (0E,0L, 2M,2H)	91.7% (0E,0L, 0M,1H)	41.7% (0E,2L, 3M,2H)
**7**	75% (0E,2L, 1M,0H)	75% (0E,2L, 0M,0H)	100%	91.7% (0E,0L, 0M,1H)	91.7% (0E,1L, 0M,0H)	91.7% (0E,1L, 0M,0H)
**8**	66.7% (0E,0L, 3M,1H)	66.7% (0E,1L, 2M,1H)	83.3% (0E,1L, 0M,1H)	83.3% (0E,0L, 1M,1H)	83.3% (0E,1L, 0M,1H)	75% (0E,1L, 2M,0H)
**9**	100%	91.7% (0E,0L, 1M,0H)	100%	100%	91.7% (0E,0L, 1M,0H)	100%
**10**	66.7% (0E,1L, 1M,2H)	91.7% (0E,0L, 0M,1H)	75% (0E,1L, 1M,1H)	83.3% (0E,1L, 0M,1H)	91.7% (0E,0L, 1M,0H)	91.7% (0E,0L, 0M,1H)
**11**	91.7% (0E,1L, 0M,0H)	83.3% (0E,0L, 1M,1H)	100%	100%	91.7% (0E,0L, 1M,0H)	100%

## Data Availability

The data presented in this study are available on request from the corresponding author. The data are not publicly available due to the ethics protocol code 2022/364.

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
