# Peer review of "Experimental Evaluation of a Hybrid Sensory Feedback System for Haptic and Kinaesthetic Perception in Hand Prostheses"

_sensors, 2023, doi:10.3390/s23208492_

Round 1

Reviewer 1 Report

The authors present an interesting study on a relevant topic in a manuscript which is an appropriate lengt, logically structured, with a clear narrative. 

The paper has a descriptive tile, a suitable abstract, and aappropriate keywords. I fount the figures to be clear with the M&M and results reported along with limitations and a number of open research questions. The practical managerial significance is addressed in the manuscript.

In summary, this is a paper that will be of interest to the intended audience and subject to the normal proofing process may be acceptrd for publication.

Reviewer 2 Report

Comments to the authors

The manuscript titled “Experimental Evaluation of a Hybrid Sensory Feedback System for Haptic and Kinaesthetic Perception in Hand Prostheses” by Sariyildiz et al. presented a new hybrid multi-modal sensory feedback system for prosthetic hands. This work is inclusive, however, it needs minor revision before it can be accepted for publication in “Sensors”.

Minor revision

1.      The difference between this study and other similar studies should be clearly stated in the introduction.

2.      F1 and F2 should be shown in Figure 2.

3.      A short video of the experiments could be added to the manuscript for better understanding.

4.      The Sensory Feedback System studies, the following articles need to be cited for better understanding in the introduction part:

https://doi.org/10.3390/prosthesis3040037

https://doi.org/10.1002/adfm.202112490.

Reviewer 3 Report

Please find an attached file.

It is fine.

Round 2

Reviewer 3 Report

The authors addressed most of the reviewers' concerns.